# Management Considerations for Pulmonary Arterial Hypertension Pharmacotherapy in the Intensive Care Unit

**DOI:** 10.3390/pharmacy11050145

**Published:** 2023-09-13

**Authors:** Elizabeth M. Foster, Danine Sullinger, James C. Coons

**Affiliations:** UPMC Presbyterian Shadyside Hospital, 200 Lothrop Street, Pittsburgh, PA 15213, USA; fosterem4@outlook.com (E.M.F.); coonsjc@upmc.edu (J.C.C.)

**Keywords:** pulmonary arterial hypertension, intensive care unit, enteral administration

## Abstract

Pulmonary arterial hypertension is a rare and progressive disease with significant morbidity and mortality risk. Several medications targeting three major disease pathways are approved for treatment. However, the management of pulmonary arterial hypertension pharmacotherapies in a patient admitted to an intensive care unit poses unique challenges. Factors such as intubation and altered mental status may prevent the continuation of home oral and/or inhaled therapy, and the progression of the disease may require escalation of therapy. This review will focus on practical management strategies for the continuation of home pulmonary arterial hypertension pharmacotherapy and escalation of therapy.

## 1. Introduction

Pulmonary arterial hypertension (PAH) is a complex disease requiring early treatment and close monitoring. Over the past three decades, an increase in the quantity and accessibility of targeted pharmacotherapy resulted in providers caring for more patients taking medications for PAH. This article will review common considerations when oral or inhaled administration of PAH medications will be managed in the acute setting. These considerations include administration via enteral access, renal and hepatic dysfunction, and common drug interactions.

PAH is a rare and progressive disease defined by a mean pulmonary arterial pressure > 20 mmHg, pulmonary capillary wedge pressure ≤ 15 mm Hg, and a pulmonary vascular resistance ≥ 2 Wood units—measurements obtained via a right heart catheterization [1]. PAH, categorized as World Health Organization (WHO) Group 1 pulmonary hypertension, is associated with many underlying etiologies, including idiopathic, heritable, drug and toxin-induced, connective tissue disorders, human immunodeficiency virus infection, and portal hypertension. The pathophysiology of PAH is complex and multifactorial, with vasculopathy being a hallmark feature [1]. When left untreated, PAH ultimately leads to right ventricular (RV) failure and death [2]. PAH has a prevalence of approximately 55 cases per million adults in developed countries [1]. Advances in pharmacotherapy over the past three decades have led to the approval of sixteen unique medications or medication formulations for patients with PAH, including phosphodiesterase-type 5 inhibitors (PDE5i), soluble guanylate cyclase (sGC) stimulators, endothelin receptor antagonists (ERAs), prostacyclin analogues (PCAs), and selective prostacyclin receptor (IP) agonists [2,3]. Combination therapy with these medications based on patient risk stratification has resulted in improvements in exercise capacity, hemodynamics, and clinical outcomes [4,5].

Despite the evolution of PAH pharmacotherapy, patients often deteriorate over time, leading to more hospitalizations and invasive approaches, including lung transplantation [6]. Patients admitted to the hospital with PAH have an in-hospital mortality rate of approximately 9%, a statistic which increases to 14–17% in patients with RV failure [7]. Patients requiring an intensive care unit (ICU) admission for acute PAH decompensation have an in-hospital mortality rate of 30–40%, which further increases in those requiring high doses of vasopressors and inotropes. Death in these patients is often caused by progressive RV failure resulting in cardiac arrest [6,7].

Patients admitted to the ICU, whether due to worsening PAH or exacerbations of unrelated disease states, possess unique clinical characteristics that must be considered by the healthcare team. In addition to optimizing volume status, providing RV support, and preserving cardiac output (CO), providers must consider how to address the patient’s PAH pharmacotherapy regimen. When reviewing home PAH medications, the team must consider the risks and benefits of continuing or modifying the regimen versus transitioning to other therapies or escalation. In the event of altered mental status or mechanical ventilation, the team should determine if medications can be given enterally, converted to parenteral formulations, or held. PAH requiring ICU care is often complicated by end-organ dysfunction, such as renal or hepatic impairment, which can be an added challenge to the management of PAH pharmacotherapy. Therefore, the focus of this review is on the management of PAH pharmacotherapy in patients who require ICU care. Specifically, this review will include recommendations for the management of chronic PAH therapies in patients who are admitted to the ICU without disease progression, as well as treatment escalation in patients presenting with disease progression.

## 2. Management of Chronic PAH Therapies in the ICU

Management of home oral and inhaled PAH therapies presents a clinical challenge in critically ill hospitalized patients. Many patients admitted to the ICU are unable to continue to take oral medications. Barriers to oral administration include mechanical ventilation, sedation, altered mental status, vomiting, malabsorption, and ileus. Many patients who are unable to swallow tablets are still able to receive medications via enteral feeding tubes. In general, we recommend the continuation of PAH therapies via enteral administration in patients unable to continue oral therapy while in the ICU. Some PAH therapies are available as a liquid formulation, while others may be crushed and mixed with liquid for enteral administration. However, many oral PAH formulations are not compatible with administration via enteral feeding tubes, such as extended-release formulations or those with special handling precautions due to teratogenicity risks [8,9,10,11,12]. In patients with concomitant gut absorption concerns, whether due to progressive right heart failure or an unrelated disease state, clinicians should be aware of the potential for variations in therapeutic effect due to malabsorption of oral therapies. In general, we recommend the continuation of home therapies with escalation to parenteral therapies if needed for progressive disease. Other patients may not have enteral access altogether. There are no established guidelines or best practices for the management of oral or inhaled PAH therapies in situations in which an interruption of oral therapies is required. Therefore, it is important to identify potential solutions to effectively manage PAH therapies in critically ill patients to avoid contributing to worsening PAH and RV failure. Clinical recommendations are provided for each medication class based on available evidence, pharmacokinetics, and other practical considerations.

### 2.1. PDE-5 Inhibitors

Sildenafil and tadalafil exert their pulmonary vasodilatory properties via the inhibition of PDE-5 receptors in the smooth muscle of the pulmonary vasculature, resulting in reduced degradation of cyclic guanosine monophosphate (cGMP) and ultimately pulmonary vascular relaxation. Sildenafil has a half-life of 4 h and is dosed three times daily, whereas tadalafil has a half-life of 35 h and is dosed once daily [13,14]. Sudden discontinuation of sildenafil is associated with clinical decompensation and RV failure [15]. Therefore, we recommend continuation of PDE5i therapy during ICU admission in the hemodynamically stable patient. In the case of oral therapy interruption, both sildenafil and tadalafil are available as oral solutions that can be given via an enteral feeding tube [13,14]. Crushing oral tablets is not addressed in package inserts for either sildenafil or tadalafil, but reports exist using sildenafil crushed and suspended in 5 mL of sterile water [16,17,18]. In patients taking tadalafil who cannot continue oral therapy, therapy may be continued using oral suspension, or the patient may be switched to sildenafil (Table 1). In general, 40 mg daily of tadalafil is equivalent to 20 mg three times daily of sildenafil. If enteral access is unavailable, sildenafil is available as an intravenous (IV) solution. Sildenafil has an oral bioavailability of 50%; therefore, patients taking 20 mg TID oral tablets should receive 10 mg TID IV solution when converting to IV administration [13]. Of note, IV therapy is significantly more costly than oral therapy; the average wholesale price of sildenafil is about $19 per 20 mg tablet and about $290 per 10 mg IV solution [19].

Both sildenafil and tadalafil undergo hepatic metabolism. Sildenafil administration in patients with mild to moderate hepatic impairment results in an 84% increase in the area under the curve (AUC) and a 47% increase in the maximum serum concentration (C_max_); sildenafil has not been studied in patients with severe hepatic impairment [13]. Tadalafil administration in mild to moderate hepatic impairment results in a similar AUC to healthy subjects although it, too, has not been studied in severe hepatic impairment [14]. Although renal elimination is minor for sildenafil and tadalafil, the AUC and C_max_ of both agents are approximately doubled in patients with severe renal impairment [13,14]. Of note, the package insert for tadalafil recommends avoidance of the product in severe renal impairment. Accumulation of the tadalafil metabolite methylcatechol occurs in renal impairment; the clinical relevance of metabolite accumulation is not well studied. In therapeutic doses in normal renal function, the metabolite is not pharmacologically active (Table 2) [14].

### 2.2. Soluble Guanylate Cyclase Stimulators

Riociguat enhances cGMP production via stimulation of sGC, resulting in pulmonary vasodilation. Riociguat has a half-life of 12 h and is dosed three times daily. There are no data on the safety of abrupt discontinuation or interruption in therapy for riociguat, therefore we recommend continuation of oral therapy if possible. It is available as an oral tablet; the package insert reports that tablets may be crushed for enteral administration; however, we do not recommend this due to the associated teratogenicity risk for the individual crushing and administering the medication to the patient [8,12]. In patients unable to continue home oral therapy, we recommend initiation of an alternative enteral agent or parenteral pulmonary vasodilator, depending on access (Table 1). Importantly, when transitioning from riociguat to sildenafil, the first dose of sildenafil must be at least 24 h after the last dose of riociguat [8]. When transitioning from riociguat to tadalafil, the first dose of tadalafil must be at least 48 h after the last dose of riociguat [14].

Riociguat undergoes hepatic metabolism to an active metabolite, which is further metabolized to inactive metabolites. About 40% of the metabolites and parent drug are excreted in the urine unchanged, although there is considerable variation in the proportion of metabolites to unchanged riociguat. There are limited pharmacokinetic data for use in hepatic and renal impairment. Riociguat is highly protein-bound and not likely to be affected by dialysis (Table 2) [8].

### 2.3. Endothelin Receptor Antagonists

ERAs block endothelin-1 receptors to prevent the vasoconstriction and proliferation associated with endothelin receptor activation. Ambrisentan, bosentan, and macitentan are the three oral ERAs used for the treatment of PAH. Ambrisentan differs from bosentan and macitentan in its selectivity; ambrisentan binds selectively to endothelin receptor type A, and bosentan and macitentan bind non-selectively to both endothelin receptor type A and B [9,10,11]. Additionally, macitentan is more tissue-selective and lipophilic than the other ERA agents [20]. The clinical implications of these selectivity differences are unknown, and no head-to-head trials have compared agents. Ambrisentan has a half-life of 9 h and is dosed daily [9]. Bosentan has a half-life of 5 h and is dosed twice daily [10]. Macitentan has a half-life of 16 h (active metabolite half-life is 48 h) and is dosed once daily [11]. There are limited data available on the safety of abrupt treatment discontinuation or interruption, therefore we recommend continuing oral therapy if possible. The ERAs all have teratogenic potential and tablets should not be crushed or split [9,10,11]. However, there is a soluble oral tablet formulation of bosentan, which can be dispersed in a small amount of water for enteral administration via a feeding tube [10]. Although several studies have demonstrated that transitioning between ERAs is safe and not associated with clinical or hemodynamic decline, switching agents presents a logistical challenge in the acute care setting due to the required enrollment in a Risk Evaluation and Mitigation Strategies (REMS) program [21,22,23]. In patients who cannot continue their ERA due to the inability to tolerate administration by mouth, we recommend switching to an alternative enteral agent or parenteral pulmonary vasodilator (Table 1).

Ambrisentan undergoes hepatic metabolism and is eliminated mainly via non-renal pathways. Although the pharmacokinetics of ambrisentan have not been studied in patients with hepatic disease, it is expected that these patients may have significantly increased exposure. Additionally, hepatotoxicity linked to other ERAs resulted in the product information recommendation to discontinue ambrisentan for patients with aspartate aminotransferase (AST) or alanine aminotransferase (ALT) values above five times the upper limit of normal (ULN), or above two times the ULN if the patient is also experiencing increased bilirubin or other signs and symptoms of hepatic impairment [9]. No studies have evaluated the use of ambrisentan in severe renal impairment; however, pharmacokinetics are likely not clinically impacted due to non-renal elimination. Ambrisentan is highly protein-bound and unlikely to be removed by dialysis (Table 2) [9].

Bosentan undergoes hepatic metabolism. Bosentan has two inactive metabolites and one active metabolite, which may contribute up to 20% of the therapeutic effect. Exposure to bosentan is reported to be unaltered in patients with mild hepatic impairment and significantly increased in patients with moderate hepatic impairment. No studies have evaluated use in patients with severe hepatic impairment. Bosentan demonstrated the highest risk of hepatotoxicity among the ERAs in clinical trials (11%). It should be avoided in patients with elevated aminotransferases (>3×ULN) at baseline, and aminotransferases should be closely monitored monthly during therapy. In patients developing aminotransferase elevations >3× ULN, doses should be held or adjusted as per package insert guidance. In patients developing aminotransferase elevations >8× ULN, bosentan should be permanently discontinued. Bosentan undergoes biliary elimination with very little renal elimination. In patients with severe renal impairment (creatinine clearance [CrCl] 15–30 mL/min), serum bosentan concentrations are unchanged, and metabolite concentrations are increased about 2-fold when compared to healthy subjects (Table 2) [10].

Macitentan undergoes hepatic metabolism to an active metabolite. In patients with hepatic impairment, exposure to macitentan and its active metabolite were observed to be decreased; however, this is likely not clinically relevant. Like ambrisentan, the product information recommends discontinuation of macitentan in patients who develop AST/ALT elevations >3× ULN or develop other signs of hepatic impairment. The elimination of macitentan is primarily non-renal. Exposure to macitentan is increased by 30% and 60% in moderate and severe renal impairment, respectively, although this is likely not clinically significant (Table 2) [11].

### 2.4. Prostacyclin Analogues and Selective IP Receptor Agonists

Treprostinil diolamine extended-release is an orally available PCA that directly produces pulmonary vasodilation and inhibits platelet aggregation and smooth muscle proliferation. It has a half-life of 4 h and is dosed two, or preferably three times daily. Abrupt discontinuation is not recommended due to the risk of clinical decompensation [24]. Therefore, we recommend continuation of therapy, whether oral, enteral, or parenteral. Treprostinil oral tablets should not be crushed or split for enteral feeding tube administration. In patients unable to continue home oral tablets, we recommend switching to an alternative enteral agent or conversion to a parenteral formulation (Table 1). Chakinala et al. reported that for a 70 kg patient, 1 mg TID of oral treprostinil has a similar pharmacokinetic profile to a 6 ng/kg/min infusion [25]. The product information also provides an equation for determining parenteral treprostinil dose requirement based on oral treprostinil daily dose [24]. When converting to parenteral treprostinil, clinical discretion should be used. We suggest using this conversion to identify an approximate target dose but starting at a lower dose and titrating to tolerability and clinical response.

Treprostinil undergoes hepatic metabolism to inactive metabolites. Treprostinil was studied in patients with mild, moderate, and severe hepatic impairment and found to have an approximately 2-fold, 5-fold, and 8-fold increased exposure, respectively. Treprostinil metabolites are eliminated renally, and very little parent drug is excreted in the urine unchanged. Patients with severe renal impairment requiring dialysis were observed to have similar exposure to treprostinil after one dose compared to healthy volunteers. Additionally, treprostinil is highly protein-bound and therefore not expected to be removed by dialysis (Table 2) [24].

Treprostinil and iloprost are available as inhaled formulations [26,27,28]. Patients in the ICU are unlikely to be able to continue these therapies in the case of intubation or altered mental status, due to a lack of evidence for the administration of these medications via alternative devices such as endotracheal tubes. In this case, we recommend switching to an alternative therapy (Table 1). It is reasonable to consider initiation of an enteral agent such as sildenafil or conversion to an alternative formulation of treprostinil.

Selexipag is a selective IP receptor agonist that is available as an oral tablet and is typically dosed twice daily. Selexipag should not be crushed for enteral administration via a feeding tube but is available as an intravenous formulation for patients who are temporarily unable to take oral therapy [29]. A post hoc analysis of the GRIPHON study found that treatment interruptions were not associated with a risk of acute clinical deterioration [30]. In patients with a treatment interruption of less than three days duration, most patients were restarted at their previous dose, while patients with a treatment interruption duration of three days or more were started at about 40% of their previous dose [30]. Based on these findings, temporary oral treatment interruption is likely safe; however, most interruptions in this study were brief, with about half of interruptions lasting less than three days. We recommend conversion to an alternate pulmonary vasodilator or IV selexipag in most cases of oral treatment interruption, especially if interruptions are expected to be more than three days in duration (Table 1). For the conversion from oral to IV selexipag, the dose should be increased by 12.5% [29]. For example, 400 mcg of oral selexipag is equivalent to 450 mcg of IV selexipag. In the case of treatment interruption, the duration of interruption should guide the dosing regimen upon restarting treatment.

Selexipag is hydrolyzed to its active metabolite by carboxylesterase before undergoing glucuronidation to inactive metabolites. Patients with mild hepatic impairment were observed to have similar selexipag exposure when compared to healthy volunteers, and patients with moderate hepatic impairment were observed to have two-fold increased exposure. Selexipag has not been studied in severe hepatic impairment. Selexipag undergoes mainly non-renal elimination. The metabolites are excreted primarily through the feces, with only 12% excreted in the urine. Patients with severe renal impairment (CrCl < 30 mL/min) were observed to have a 40–70% increased AUC compared to healthy volunteers. Selexipag has not been studied in patients with CrCl < 15 mL/min or undergoing dialysis. Selexipag and its active metabolite are highly protein-bound and unlikely to be removed by dialysis (Table 2) [29].

## 3. Escalation of Therapy

In patients with decompensated PAH who are admitted to the ICU, escalation of therapy may be indicated. In addition to treatment escalation, supportive care should be initiated and the cause of deterioration should be addressed [3]. Common treatable triggers include infection, arrhythmias, anemia, thromboembolic disease, pericardial effusion, metabolic abnormalities, and exacerbations of other chronic comorbidities [1,3]. Supportive care measures include monitoring hemodynamic parameters with a focus on optimizing volume status, maintaining blood pressure and mean arterial pressure (MAP), maintaining CO, and reducing pulmonary afterload [6]. Utilization of vasopressors, vasodilators, and inotropes should be considered to maintain these parameters although are not discussed extensively in this review [1]. In general, inotropic therapy should be used to improve RV contractility and thus stroke volume. Vasopressors are often required in combination with inotropes to maintain adequate blood pressure and perfusion. Vasopressin may have a pulmonary vasodilatory effect in addition to systemic vasoconstriction so may be preferred in this patient population. Norepinephrine may lead to pulmonary vasoconstriction but appears to improve RV-arterial coupling due to β_1_ agonistic properties [3]. The team should also focus on reducing tachycardias and avoiding both renal and hepatic injury [6]. Hypoxia is a common challenge that further exacerbates elevated pulmonary pressures. Supplemental oxygenation should be used in addition to optimizing pharmacotherapies to avoid intubation if possible. Intubation leads to decreased systemic vascular resistance (SVR) and increased transpulmonary pressures, which can further decrease CO. Additionally, sedative medications used peri- and post-intubation can depress cardiac function [31]. In institutions with the appropriate resources, extracorporeal life support (ECLS) could be considered. This approach often allows for the continuation of both oral and inhaled maintenance medications.

In the setting of acute decompensated PAH, home medications should be managed as described above. In the event that maintenance medications are unable to be continued or adjusted, or if there is minimal response to adjustments and substitutions along with supportive care, escalation may be required to support the patient. Escalation of pharmacotherapy includes initiation of IV prostacyclin analogues and inhaled pulmonary vasodilators.

### 3.1. IV Prostacyclin Analogues

In high-risk treatment-naive patients, or patients with disease progression on enteral or inhaled therapies, IV prostacyclin analogues are indicated. Epoprostenol is the preferred agent due to potent pulmonary vasodilation with a rapid onset, short half-life (<6 min) allowing quick titrations, pronounced afterload reduction, and overall reduction in mortality [3,6,7,31,32]. In patients with chronic severe PAH, IV prostacyclin analogues are highly effective and associated with a survival benefit [6,7,32]. Early initiation of IV epoprostenol is specifically recommended in high-risk patients with RV failure due to PAH who are not already receiving IV PCA therapy. Epoprostenol should be initiated at 1–2 ng/kg/min and titrated rapidly in the ICU to a maximally tolerated dose, while avoiding hypotension and worsening ventilation-perfusion (V/Q) mismatch. Other common adverse effects include nausea, vomiting, flushing, jaw or musculoskeletal pain, diarrhea, headache, and thrombocytopenia [7,31]. Slower titration in addition to supportive medications may be needed to avoid such adverse effects.

Treprostinil is an alternative prostacyclin analogue that may be considered. Treprostinil is available as a continuous IV or subcutaneous (SQ) infusion, but IV is preferred in critically ill patients due to concern for edema, which may delay or limit SQ absorption [1,32]. If worsening hypoxemia occurs after IV PCA initiation due to V/Q mismatch or development of pulmonary edema, the team should consider etiologies other than PAH.

### 3.2. Inhaled Therapies

In patients experiencing clinical decompensation on IV PCA therapy, inhaled pulmonary vasodilators could be considered. Inhaled pulmonary vasodilators locally dilate pulmonary arterioles in ventilated lung units, improving oxygenation via V/Q matching and reducing PVR [7,33]. These agents can also reduce inflammation and offer protective effects on cells while avoiding systemic vasodilation and hypotension associated with IV therapies [33]. Inhaled therapies are used to treat severe sustained PAH and intractable hypoxia unresponsive to supplemental oxygen after initiation of maximally tolerated IV therapies [32]. The current body of literature describing the use of combination inhaled and intravenous prostacyclin analogues includes mostly non-Group 1 PH patients requiring escalating therapies due to decompensated RV failure or in the peri-operative setting. Although this strategy requires further exploration within the PAH population, inhaled therapies could be a temporizing measure while IV PCA is titrated and supportive care is initiated [7,33]. Inhaled pulmonary vasodilators include nitric oxide and aerosolized prostacyclins or analogues, such as epoprostenol, iloprost, and treprostinil. Inhaled epoprostenol is the preferred option in most institutions, due to widespread availability and lower cost compared to other therapies, such as nitric oxide. However, favorable outcomes were described with inhaled treprostinil therapy, so initiation may be considered in some patients [33]. As previously mentioned, inhaled treprostinil has not been shown to be reliably delivered in patients who are mechanically ventilated or who have altered mental status. The lack of supporting literature, cost, and availability of this medication along with the required administration device can be a significant limiting factor.

In patients who were transitioned to parenteral therapy due to lack of enteral access, home oral therapies should be restarted once oral access is available. Parenteral therapy should be weaned slowly as oral therapies are initiated, and abrupt discontinuation should be avoided due to the risk of rebound PAH symptoms [31].

## 4. Conclusions

The management of PAH pharmacotherapy is often complicated in the ICU. Factors such as intubation and altered mental status may affect the ability of the patient to continue home oral medications. Also, the progression of the disease may require the escalation of PAH therapies. A nuanced approach to managing these situations in critically ill patients with PAH is needed.

## Figures and Tables

**Table 1 pharmacy-11-00145-t001:** Management recommendations for PAH pharmacotherapy in the setting of oral or inhaled treatment interruption.

Medication	Enteral Administration Options	Parenteral Administration Options	Management Recommendation for Oral/Inhaled Treatment Interruption
Ambrisentan	Oral tablet	N/A	Enteral access: oral tablet may not be crushed; switch to alternative enteral agent, such as sildenafil. Parenteral access: initiate alternative pulmonary vasodilators, such as IV sildenafil or epoprostenol.
Bosentan	Oral tablet Soluble oral tablet	N/A	Enteral access: continue therapy if soluble oral tablet is available; if soluble oral tablet unavailable, switch to alternative enteral agent, such as sildenafil. Parenteral access: initiate alternative pulmonary vasodilators, such as IV sildenafil or epoprostenol.
Macitentan	Oral tablet	N/A	Enteral access: oral tablet may not be crushed; switch to alternative enteral agent, such as sildenafil. Parenteral access: initiate alternative pulmonary vasodilators, such as IV sildenafil or epoprostenol.
Sildenafil	Oral tablet Oral suspension	Intravenous	Enteral access: continue therapy with oral suspension or crushed tablet in 5 mL sterile water. Parenteral access: convert to IV sildenafil (1:2 IV:PO conversion).
Tadalafil	Oral tablet Oral suspension	N/A	Enteral access: continue therapy with oral suspension or switch to sildenafil suspension/crushed tablet in 5 mL sterile water (40 mg daily tadalafil = 20 mg TID sildenafil). Parenteral access: convert to IV sildenafil.
Treprostinil	Oral tablet	Intravenous Inhalation Subcutaneous	Enteral access: oral tablet may not be crushed; convert to enteral agent, such as sildenafil. Parenteral access: convert to parenteral treprostinil (in 70 kg patient, 1 mg TID PO = 6 ng/kg/min).
Iloprost	N/A	Inhalation	Convert to alternate agent if unable to continue inhaled therapy.
Selexipag	Oral tablet	Intravenous	Short-term (<3 days) treatment interruption likely safe. Enteral access: oral tablet may not be crushed; switch to alternative enteral therapy, such as sildenafil. Parenteral access: convert to IV selexipag (Table 1).
Riociguat	Oral tablet	N/A	Enteral access: oral tablets should not be crushed; switch to alternative enteral therapy, such as sildenafil (must delay starting sildenafil for 24 h after last riociguat dose). Parenteral access: initiate alternative pulmonary vasodilators, such as IV sildenafil or epoprostenol.

IV = intravenous; PO = by mouth; TID = three times daily.

**Table 2 pharmacy-11-00145-t002:** Pharmacokinetic characteristics and considerations for PAH oral and inhaled pharmacotherapies.

Medication	Metabolism	Elimination	Recommendations in Renal Failure	Recommendations in Hepatic Failure	Notable Drug–Drug Interactions
Ambrisentan	Hepatic via CYP3A4, CYP2C19, and UGT1A9S, 2B7S, and IA3S	Nonrenal	No dose adjustments	Avoid in moderate to severe hepatic impairment Discontinue if signs of liver injury	CYP3A4 and P-gp inhibition by cyclosporine results in increased ambrisentan exposure. Recommend maximum ambrisentan dose of 5 mg daily when co-administered with cyclosporine.
Bosentan	Hepatic via CYP2C9 and CYP3A4	Nonrenal	No dose adjustments	Avoid in moderate to severe hepatic impairment Discontinue if signs of liver injury	Moderate to strong CYP3A4 inhibitors (e.g., ritonavir, ketoconazole, cyclosporine) and CYP2C9 inhibitors (e.g., fluconazole, amiodarone) increase bosentan exposure. Recommend avoidance or dose reduction. Avoid co-administration with combinations of CYP2C9 and CYP3A4 inhibitors. CYP2C9 induction by bosentan may result in decreased HMG CoA reductase levels. Recommend monitoring. CYP3A4 induction by bosentan may reduce levels of CYP3A4 metabolized medications (e.g., cyclosporine, hormonal contraceptives). Avoid use or monitor closely as appropriate. Glyburide and cyclosporine contraindicated with bosentan due to hepatotoxicity risk.
Macitentan	Hepatic via CYP3A4 (major), CYP2C8, CYP2C9, and CYP2C19 (minor) Metabolized to active metabolite	Nonrenal	No dose adjustments	No dose adjustments Discontinue if signs of liver injury	Moderate to strong CYP3A4 inhibitors (e.g., ritonavir, ketoconazole, cyclosporine) and CYP2C9 inhibitors (e.g., fluconazole, amiodarone) increase macitentan exposure. Recommend monitoring, dose reduction, or avoidance. Avoid co-administration with combinations of CYP2C9 and CYP3A4 inhibitors. Strong CYP3A4 inducers (e.g., rifampin) reduce macitentan levels. Avoid concomitant use.
Sildenafil	Hepatic via CYP3A4 (major) and CYP2C9 (minor) Metabolized to active metabolites (50% potency)	Feces, urine (13%)	Consider dose reduction if CrCl < 30 mL/min or HD Monitor for adverse effects	Avoid in severe hepatic impairment	Concomitant nitrates and riociguat contraindicated. Strong CYP3A4 inhibitors (e.g., ritonavir, ketoconazole) increase sildenafil/tadalafil exposure—avoid use. Moderate to strong CYP3A4 inducers (e.g., bosentan, phenytoin, rifampin) reduce sildenafil/tadalafil exposure- may need increased dose. Recommend dose reduction when discontinuing concomitant CYP3A4 inducer.
Tadalafil	Hepatic via CYP3A4	Feces, urine (36%)	Avoid if CrCl < 30 mL/min or HD	Avoid in severe hepatic impairment	
Treprostinil	Hepatic via CYP2C8 (major) and CYP2C9 (minor)	Urine (mostly inactive metabolites, only 4% unchanged)	No dose adjustments	Contraindicated in severe hepatic impairment	Strong CYP2C8 inhibition by gemfibrozil results in increased treprostinil exposure, recommend starting dose reduction to 0.125 mg BID.
Iloprost	Hepatic via beta-oxidation of carboxyl side chain	Urine (68%), feces	No dose adjustments	No dose adjustments	No notable CYP metabolic interactions.
Selexipag	Hydrolyzed to active metabolite by carboxylesterase Hepatic via CYP2C8 (major) and CYP3A4 (minor), UGT1A3 and UGT2B7	Feces, urine (12%)	Avoid if CrCl <15 or HD	Decrease frequency to once daily in moderate hepatic impairment Avoid in severe hepatic impairment	Strong CYP2C8 inhibitors (e.g., gemfibrozil) increase selexipag exposure—avoid use. Moderate CYP2C8 inhibitors (e.g., clopidogrel, leflunomide) increase selexipag exposure—recommend dose reduction to once daily. CYP2C8 inducers (e.g., rifampin) decrease selexipag exposure, recommend increasing dose when initiating interacting medication and dose reduction if discontinuing interacting medication.
Riociguat	Hepatic via CYP1A1, CYP3A4, CYP3A5, and CYP2J2	Urine	Avoid if CrCL < 15 mL/min or HD	Avoid in severe hepatic impairment	Concomitant nitrates and PDE-5 inhibitors contraindicated. Separate from antiacid medication by at least 1 h due to decreased absorption at increased pH. Strong CYP3A and P-gp/BCRP inhibitors (e.g., ritonavir, ketoconazole, itraconazole) increase riociguat exposure—recommend a reduction in starting dose to 0.5 mg TID and monitoring for hypotension. Strong CYP3A inducers (e.g., phenytoin, rifampin, carbamazepine) reduce riociguat exposure. No dosing recommendations exist. Tobacco smoke is a CYP1A1 inducer, resulting in reduced riociguat exposure. In patients on increased doses of riociguat due to smoking, dose reduction may be prudent in admitted patients who are no longer smoking.

CrCl = creatinine clearance; CYP = cytochrome P450; HD = hemodialysis; P-gp = P-glycoprotein; BCRP = breast cancer resistance protein.

## Data Availability

No new data were created or analyzed in this study. Data sharing is not applicable to this article.

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
