# Peer review of "Management Considerations for Pulmonary Arterial Hypertension Pharmacotherapy in the Intensive Care Unit"

_pharmacy, 2023, doi:10.3390/pharmacy11050145_

Round 1

Reviewer 1 Report

This is a useful review but would benefit from moving the outline of the scope of the manuscript earlier in the introduction and the title is misleading.

ICU patients with PAH generally have severe RHF however there is nothing in the manuscript describing management, merely a statement that it needs to be managed. Perhaps the title then should reflect the true content better such as "Suggested approaches towards delivering PAH targeted therapy in patients admitted to the ICU"

Whilst renal and hepatic dysfunction are discussed, the potential gut problems in such patients are not and are of particular relevance to planned continuation of oral therapies, particularly in patients with RHF.

Problems due to mechanical ventilation are described but clinical practice is moving toward ECMO support for decompensated patients which allows oral medicating. Worth a sentence or two on this relevant point.

Drug interactions are very important and nowhere more relevant than in patients ventilated on the ITU, I think the manuscript may benefit from a section drawing the readers attention to this eg antifungal therapies and some antibiotics of relevance to patients on ICU with sepsis. 

Author Response

Thank you so much for the feedback - the majority of the suggested edits were incorporated into the updated manuscript.

Reviewer 2 Report

The submission, as a Review, is well organized and is interesting to read.

It has a valuable application in the clinical field and can also be used for educational purposes.

Only minor editing is recommended to eliminate some duplications in the text or in the explanations.

The goal is to make paragraphs shorter and to integrate the information that could lead straight forward into conclusions.

The submission is well organized and English is well used.

Author Response

Thank you for the feedback - the authors reviewed the manuscript and attempted to remove duplications.

Reviewer 3 Report

This is a review focused on the management of pulmonary arterial hypertension (PAH) pharmacotherapy in patients admitted to the ICU due to worsening PAH or unrelated acute disease states, situations where discontinuation of chronic oral therapies is required or an escalation of treatment is needed.

In the absence of established guidelines or best practices for managing PAH therapies in such situations, the authors provide an interesting and comprehensive overview of currently available oral and inhaled PAH therapies, the availability of enteral and/or parenteral delivery options, management recommendations aimed at avoiding interruption of therapy leading to clinical decompensation and right ventricular failure, recommendations in case of renal and/or hepatic insufficiency based on pharmacokinetic characteristics, modalities of intensification of pharmacotherapy of PAH; all this with a practical and applicable approach.

I have no particular changes to suggest to the authors, except for a small formal revision of the text and in the case of Selexipag (lines 231-257) the addition of the daily dose of the drug, in analogy to what was done with the previous drugs.

Author Response

Thank you for the feedback - the selexipeg edits have been incorporated